# TBCC Domain-Containing Protein Regulates Sporulation and Virulence of *Phytophthora capsici* via Nutrient-Responsive Signaling

**DOI:** 10.3390/ijms252212301

**Published:** 2024-11-16

**Authors:** Yaru Guo, Xiang Qiu, Bingting Lai, Caihuan Ou, Huirong Wang, Hengyuan Guo, Linying Li, Lili Lin, Dan Yu, Wenbo Liu, Justice Norvienyeku

**Affiliations:** 1Key Laboratory of Green Prevention and Control of Tropical Diseases and Pests, Ministry of Education, School of Tropical Agriculture and Forestry, Hainan University, Haikou 570228, China; guoyaru0328@163.com (Y.G.); 18889571791@163.com (X.Q.); ice_stop@hainanu.edu.cn (B.L.); 17889860838@163.com (C.O.); 17889984858@163.com (H.W.); guohengyuan0428@163.com (H.G.); lly3230239731@163.com (L.L.); yudan37@westlake.edu.cn (D.Y.); saucher@hainanu.edu.cn (W.L.); 2Ministerial and Provincial Joint Innovation Centre for Safety Production of Cross-Strait Crops, Fujian Agriculture and Forestry University, Fuzhou 350002, China; lilly116@163.com

**Keywords:** *Phytophthora capsici*, tubulin-binding cofactor C, pathogenesis, plant–pathogen interactions, encystment, chloroplast-targeting peptide

## Abstract

Phytopathogenic oomycetes, particularly *Phytophthora capsici*, the causal agent of Phytophthora blight disease in essential vegetables and fruit crops, remains a persistent challenge in the vegetable production industry. However, the core molecular regulators of the pathophysiology and broad-range host characteristics of *P. capsici* remain unknown. Here, we used transcriptomics and CRISPR-Cas9 technology to functionally characterize the contributions of a novel gene (*PcTBCC1*) coding for a hypothetical protein with a tubulin-binding cofactor C domain with a putative chloroplast-targeting peptide (cTP) to the pathophysiological development of *P. capsici*. We observed significant upregulation in the expression of *PcTBCC1* during pathogen–host interactions. However, the vegetative growth of the ∆*Pctbcc1* strains was not significantly different from the wild-type strains. *PcTBCC1* gene replacement significantly compromised the sporulation, pathogenic differentiation, and virulence of *P. capsici*. At the same time, ∆*Pctbcc1* strains were sensitive to cell wall stress-inducing osmolytes. These observations, coupled with the close evolutionary ties between *PcTBCC1* and pathogenic oomycetes and algae, partly support the notion that *PcTBCC1* is a conserved determinant of pathogenesis. This study provides insights into the significance of tubulin-binding cofactors in *P. capsici* and underscores the potential of PcTbcc1 as a durable target for developing anti-oomycides to control phytopathogenic oomycetes.

## 1. Introduction

Tubulin is a cytoskeletal polymer formed by the assemblage of heterodimers, including α-tubulin and β-tubulin. Tubulin multi-protein superfamily members comprise functionally distinct subtypes: a, *b*, *g*, *d*, *e*, *z*, *h*, *q*, and *t.* Based on their inherent functional domains, *α*-tubulin and β-tubulin are classified as N-terminal, middle, and C-terminal *a/b* tubulin domain proteins [1]. Additional studies have reclassified C-terminal tubulin domain-containing proteins into six subgroups based on the amino acid composition of the conserved functional domain, with the amino acid sequences of C-terminal α- and β-tubulin domain from the same species sharing a high identity. For instance, studies showed that sequences of the C-terminal α- and β-tubulin domain proteins from mice shared up to 91–100% [2]. In addition, the terminal structure interacts with tubulin-related protein MAPs to regulate tubulin assembly [3].

Mechanistically, some chaperones and protein cofactors play significant roles in modulating the proper folding of α/β-tubulin heterodimers. Studies have shown that stable interactions between the tubulin-binding cofactor (TBC) and synthesized polypeptides of tubulin bonded to cytosolic chaperones facilitate the proper folding of tubulin proteins [4]. Tubulin binding cofactor C (TBCC) is an essential protein for the correct folding of α- and β-tubulin and their polymerization into microtubule heterodimers [5].

Research demonstrations have shown that the five types of tubulin-binding factors (TBCS) modulate the progression of vital cellular and physiological processes by exerting direct regulatory roles in tubulin synthesis and the polymerization of microtubules [6]. TBCC dysfunction triggers the development of neurological diseases [7]. Additional research has shown that the TBCC-dependent regulation of GTPase activity and the mediation of native tubulin folding exert significant inhibitory effects on tumor and breast cancer cells [8,9]. In humans, tubulin-binding cofactor C (TBCC) represents a post-chaperone binding protein that essentially promotes the proper folding and assemblage of α- and β-tubulin monomers and the efficient formation of tubulin heterodimers and results in the regulation of microtubule polymerization [10]. NMR spectroscopy-based research investigations revealed the location of TBCC at the centrosome and also confirmed the existence of physical interactions between dimers of N-terminal domain-containing α/β-tubulin proteins and TBCC [11]. The subsequent treatment of HeLa cell cultures with TBCC RNAi revealed a significant depletion in TBCC, resulting in multipolar spindles and mitotic failure, underscoring the role of TBCC in bipolar spindle formation [12]. The tubulin-specific chaperone cofactor C regulates retinitis pigmentosa 2 [10].

Although the impact of tubulin-mediated regulation of cellular processes, including trafficking, cell division, cell morphogenesis, organelle positioning, cell migration, and polarity, on diverse developmental transformations in different organisms has been demonstrated [13], the direct and indirect influence of TBC and tubulin on the morphological and pathological development of *P. capsici* and other economically crucial phytopathogenic oomycetes are still unknown.

In the current study, we identified nine TBC domain-containing proteins in *P*. *capsici,* the causal agent of foliar blight, damping-off, wilting, and root, stem, and fruit rot disease in more 45 fruits and vegetable crops—including pepper (*Capsicum annum*), tomato (*Solanum lycopersicum*), kidney bean (*Phaseolus vulgaris*), eggplant (*Solanum melongena)*, faba bean (*Vicia faba*), watermelon, (*Citrullus lanatus*), and squash (*Cucurbita moschata)*, among others. We monitored the transcriptomic dynamics of TBC-coding genes during the early and late stages of *P. capsici* interaction with six susceptible hosts, including pepper, tomato, squash, faba bean, kidney bean, and watermelon. These examinations revealed the exclusive upregulation of the expression of three TBCC domain-containing proteins, namely, *PcTBCC1*(DVH05_025197), *PcTBCA* (DVH05_026725), and *PcTBCE* (DVH05_001436), in the early and late stages of *P. capsici*–host interactions. The phenotypic and biochemical characterization of *PcTBCC*1 gene-defective strains generated in this study using the integrated application of homologous recombination and CRISPR-Cas9 techniques revealed a novel PcTbcc1 domain-containing hypothetical protein that significantly contributes to the reproduction and pathogenesis of *P*. *capsici.*

## 2. Results

### 2.1. Identification, Phylogeny, and Transcriptomic Dynamics of Genes Coding for TBCC Domain-Containing Proteins in P. capsici

The nine tubulin-binding cofactor C domain-containing proteins in *P. capsici* were identified by performing BLASTp and reverse-BLASTp search analyses with amino acid sequences of TBC domain-containing proteins retrieved from the Ensembl Genomes (http://ensemblgenomes.org/, accessed on 09 February 2022) database [14] for *Schizosaccharomyces pombe*, *Neurospora crassa*, *Aspergillus nidulans*, *Magnaporthe oryzae*, *Chlamydomonas reinhardtii*, *Styela clava*, *Stegodyphus dumicola*, *Breviolum minutum*, *Capsicum annuum*, *Volvox carteri f. nagariensis*, *Arabidopsis thaliana*, *Aureococcus anophagefferens*, *Saprolegnia parasitica*, *Phytophthora sojae,* and *Phytophthora infestans.* Further phylogenetic examinations showed that PcTbcc shared evolutionary ties with *P. infestans*, *S. parasitica,* and *A. anophagefferens* but was evolutionarily distant from *P. sojae* (Figure 1a). Protein sequence feature analysis showed that, in addition to the conserved TBCC domain identified in this study, protein sequences from plants and fungi possess an additional TBCC_N domain. Interestingly, we also showed that while SpTbcc and AaTbcc possess nuclear-targeting peptides (nTPs), PcTbcc1 and SpaTbcc contain an exclusive chloroplast-targeting peptide (cTP) and mitochondrion-targeting peptide (mTP), respectively (Figure 1b). These results indicate a possible divergence in the functions of TBCC domain-containing proteins in oomycetes compared to plants and fungi.

We monitored the expression patterns of nine *P. capsici* TBC domain-containing coding genes during the early and late stages of pathogen–host interactions. Comparative transcriptomic profiling revealed a significant upregulation of TBCA (DVH05_026725), TBCC1 (DVH05_025197), TBCC-2 (DVH05_019376), and TBCE (DVH05_001436), with TBCC1 exhibiting the highest expression intensities in the early and late stages of *P. capsici* interactions with pepper, tomato, squash, faba bean, kidney bean, and watermelon seedlings (Figure 1c). Based on these observations, we hypothesized that TBCC1 functions as a crucial pathogenic determinant that regulates the pathogenicity and virulence of *P. capsici* in multiple hosts.

### 2.2. PcTbcc1 Functions as a Negative Genetic Regulator of Vegetative Growth in P. capsici Under Nutrient-Deficient Conditions

We examined the likely effects of PcTbcc1 dysfunctions on the vegetative development of *P. capsici* by comparing the colony diameters of the *TBCC1* targeted gene deletion strains (∆*Pctbcc1-212* and ∆*Pctbcc1-297*), TBCC domain-defective strains (*Pctbcc1-72*^∆*115-611*^ and *Pctbcc1-124*^∆*115-611*^), the strains harboring the empty pYF515 and pBS plasmids (EV), and the wild-type strains cultured on clarified V8 juice agar medium. The results showed that the complete deletion of both full-length *PcTBCC1* and genetic disruptions in the TBCC domain had no significant adverse effects on the vegetative growth of *P. capsici* under nutrient-sufficient growth conditions (Figure 2a,b).

We, however, recorded significant increases in the vegetative growth of *∆Pctbcc1-212, ∆Pctbcc1-297*, *Pctbcc1-72^∆115-611^, and Pctbcc1-124^∆115-611^* cultured on nutrient-deficient [15] medium compared to the wild-type and the EV strains (Figure 2c,d). Simultaneously, we demonstrated that PcTbcc1 dysfunction has no adverse influence on the vegetative hyphae produced by the *PcTBCC1* gene-defective strains (Figure 2e). We hypothesized that PcTbcc-1 possibly negatively regulates vegetative growth in *P. capsici* under nutrient-limited conditions.

### 2.3. PcTbcc1 Exerts Differential Influence on Stress Homeostasis in P. capsici

We evaluated the significance of PcTbcc1 to stress tolerance in *P. capsici*; the individual strains were cultured using V8 juice agar medium supplements with different stress-inducing agents, including 2 mM DL-dithiothreitol (DTT), 200 µg/mL Congo red (CR), 20 µg/mL calcofluor-white (CFW), 3% ^w^/_v_ cellulose, 2 mM hydrogen peroxide (H_2_O_2_), and 0.5 M sodium chloride (NaCl). The corresponding results obtained from stress response assays revealed significant inhibition in the vegetative growth of ∆*Pctbcc1-212*, ∆*Pctbcc1-297*, *Pctbcc1-72*^∆*115-611*^, and *Pctbcc1-124*^∆*115-611*^ strains cultured on media supplemented with CR, cellulose, and H_2_O_2_. Conversely, the addition of exogenous CFW and NaCl triggered significant changes in the vegetative growth of *P. capsici* (Figure 3a,b). From these results, we concluded that PcTbcc1 differentially modulates the response of *P. capsici* to different stress conditions.

### 2.4. Targeted Deletion of the PcTBCC1 Gene Causes Significant Attenuation in Sporangium Production

To unravel both the direct and indirect contributions of PcTbcc-1 to asexual reproduction and disease dissemination, the ∆*Pctbcc1-212*, ∆*Pctbcc1-297*, *Pctbcc1-72*^∆*115-611*^, *Pctbcc1-124*^∆*115-611*^, EV, and the wild-type strains cultured on V8 juice agar medium for 5 d under 25 °C in darkness were exposed to 12 h of alternating light and dark conditions for 3 d to induce sporangium production. Results from the assessment of sporangium production in the individual strains showed that the targeted disruption of *PcTBCC1* triggered a significant reduction in sporangium production in *P. capsici* without altering morphological stature (Figure 4). From these results, we infer that PcTbcc1 promotes the asexual reproduction and efficient dissemination of the Phytophthora blight pathogen.

### 2.5. PcTbcc1 Plays Essential Roles in Mediating Encystment, Cyst Release, and Survival of Zoospores in P. capsici

Zoospore/cyst formation, cyst release, cyst survival, and cyst germination are crucial events in the life cycle of *P. capsici*. Here, we examined the contribution of PcTbcc1 to pathogenic differentiation in the Phytophthora blight pathogen by monitoring the encystment, cyst release, and survival of zoospores. Sporangium suspensions prepared using spores harvested from the individual strains were inoculated on micro-cover slides and monitored under a microscope at 0, 30 and 60 min post-inoculation at 4 °C. These examinations showed that the targeted replacement of *PcTBCC1* severely compromised encystment, triggered the abortive release of sporangia, and attenuated the survival of zoospores produced by the *PcTBCC1* gene-defective strains (Figure 5). Therefore, we concluded that PcTbcc1 promotes the progression of pathogenic differentiation in *P. capsici* by modulating encystment, cyst release, and survival. We further speculate that PcTbcc1 positively influences zoospore survival by directly or indirectly mediating oxidative stress homeostasis in zoospores.

### 2.6. PcTbcc1 Plays a Significant Role in Promoting the Full Virulence of P. capsici

We investigated the influence of the *PcTBCC1* gene on the infection capabilities of *P. capsici* by challenging the seedling of *P. capsici*-susceptible (HNUCB0226) pepper cultivars with mycelia plugs of ∆*Pctbcc1-212*, ∆*Pctbcc1-297, Pctbcc1-72*^∆*115-611*^, *Pctbcc1-124^∆115-611^*, EV, and the wild-type strains cultured using V8 juice medium incubated under 25 °C with 12 h daylight cycle conditions for 3 days. The results revealed a significant attenuation in virulence capabilities of ∆*Pctbcc1-212*, ∆*Pctbcc1-297*, *Pctbcc1-72*^∆*115-611*^, and *Pctbcc1-124^∆115-611^*, compared to the virulence characteristics exhibited by the EV and wild-type strains (Figure 6a). Further lesion index analyses on ∆*Pctbcc1-212* and ∆*Pctbcc1-297* showed similar virulence defects between the full-length *PcTBCC-1* gene replacement (∆*Pctbcc1-212* and ∆*Pctbcc1-297*) and TBCC1 domain disruption strains (*Pctbcc1-72*^∆*115-611*^ and *Pctbcc1-124^∆115-611^*) (Figure 6b). The results suggest that the TBCC domain plays a crucial role in facilitating the optimal function of PcTbcc1 as a virulence determinant of *P. capsici.*

## 3. Discussion

Microtubules are evolutionarily conserved principal components of the cytoskeleton across eukaryotic cells. Tubulins tightly regulate structural dynamics and polymerization, define the polymerization of microtubules, and constitute the central building blocks of microtubules [16]. Microtubules play indispensable regulatory roles in vital cellular processes, including mitotic cell division, cell motility, the progression of cellular transport, pathogenesis, and the maintenance of cell structure and shape [17,18]. Microtubule dysfunction has been linked to several life-threatening diseases, including Alzheimer’s disease, hearing impairment, and cancer [19]. In the current study, transcriptomic investigations revealed significant upregulation in the expression pattern of the single-copy *P. capsici* gene *PcTBCC1*, which codes for a hypothetical protein containing the tubulin-binding cofactor C (TBCC) domain during pathogen–host interactions, indicating the potential contribution of *PcTBCC-1* to the pathogenesis of the Phytophthora blight pathogen. We demonstrated that TBCC sequences obtained from sixteen organisms possess the conserved TBCC domain and that the novel TBCC domain-containing protein identified in *P. capsici* shares closer evolutionary ties with *Aureococcus anophagefferens* (an alga pelagophyte that causes harmful brown tide blooms in marine embayments) [20], and two economic pathogenic oomycetes, *Phytophthora infestans* and *Saprolegnia parasitica* [21,22], but is evolutionarily distant from the phytopathogenic oomycetes *Phytophthora sojae* and selected fungi and plants. Based on these observations, we speculated that PcTbcc1 affects the infection characteristics of *P. capsici* by regulating unique infection-related pathways that may have been lost in some pathogenic oomycetes and fungal species due to speciation. Detailed sequence feature examinations revealed Pctbcc1 as a putative non-classically secreted extracellular protein with a chloroplast-targeting peptide (cTP) signal embedded within the TBCC domain. Thus, its secretion, the precise influence of cTP on the virulence of *P. capsici* during host–pathogen interaction, its probable roles as an effector protein, and its likely interacting partners will be the focus of subsequent investigations.

Tubulin-folding cofactors (including TBCA, TBCB, TBCC, TBCD, and TBCE) accelerate morphological development in plants, animals, and fungi. For instance, in *Fusarium asiaticum*, the targeted disruption of *FaTBCA* triggers a significant reduction in vegetative growth [23]. Additionally, the TBC-mediated regulation of cell division promotes morphological development in *Zea mays* and *Arabidopsis thaliana* [24,25,26]. In contrast, the targeted replacement of *PcTBCC1* in *P. capsici* had no adverse effects on the vegetative development of the polycyclic Phytophthora blight pathogen. From these results, we hypothesized that, unlike its counterparts in plants and fungi, the regulatory roles of PcTbcc1 in *P. capsici* likely overlap with those of other members of the tubulin-folding cofactor family. PcTbcc1 directly regulates vegetative development and assumes new roles in the pathophysiological development of *P. capsici*.

In addition, the promotion of growth, microtubules, and tubulin-folding cofactor-dependent processes (including intracellular and vesicular transport) facilitates the transport of cargoes, virulent determinants (secreted proteins/effector proteins), and the relay of internal cues necessary to drive pathogenic differentiation in invading pathogens during pathogen interactions. For instance, the targeted disruption of microtubule-associated dynein motor protein in rice blast pathogen abolished conidiation and rendered the ∆*Modync1I2* strains non-pathogenic [27]. Similarly, we demonstrated that the deletion of *PcTBCC1* significantly suppressed asexual sporulation, delayed encystment, compromised zoospore survival, and attenuated the virulence of *P. capsici.*

In addition, pathogenic microbes that cause diseases in seasonal crops experience a window of over-wintering/over-summering periods, often associated with nutrient limitation and multiple harsh conditions. Most phytopathogenic microbes have evolved diverse survival strategies (by going into either total or partial hibernation) to drastically reduce the rate of metabolism and other energy-intensive physiological developmental processes, including vegetative growth [28]. The current study showed that the targeted disruption of *PcTBCC1* caused a significant increase in the vegetative growth of *PcTBCC1* gene-defective strains under nutrient-deficient conditions. Meanwhile, the vegetative growth of ∆*Pctbcc1* strains was indistinguishable from the wild-type strains under nutrient-sufficient conditions. These observations suggest that PcTbcc1 mediates the economic utilization of nutrients to ensure the survival of *P. capsici* under starvation- or stress-prone conditions by acting directly or indirectly to minimize vegetative morphogenesis and the rate of metabolic processes. These observations are in tandem with the fundamental knowledge that fungi and oomycetes generally switch from the vegetative growth phase to asexual sporulation (i.e., the reproductive phase) in response to nutrient limitations [29,30]. Coupled with the significant reductions observed in sporulation characteristics of the ∆*Pctbcc1* strains, there is adequate support for the reasoning that PcTbcc1 functions as a nutrient-sensitive genetic switch that promotes asexual sporulation by suppressing morphological differentiation to support the survival of *P. capsici* under nutrient harsh conditions and facilitating the mobilization of nutrients from the host plant by enhancing the virulence of *P. capsici* (Figure 7).

Microtubules and tubulins are crucial targets of diverse fungicides [31,32]. However, the direct contribution of microtubules, tubulins, and tubulin-folding cofactors to the pathophysiological development of oomycetes remains unknown. Understanding the regulatory parameters of pathogenesis in this pathogen is of great concern to the broader scientific community because it poses a substantial economic burden. Findings from the current investigations underscore the crucial contributions of PcTbcc1, a novel TBCC domain-containing hypothetical protein, to the sporogenesis and virulence of *P. capsici*. These findings provide theoretical and practical insights that will facilitate the deployment of PcTbcc-1 and likely other tubulin-folding cofactors in *P. capsici* as targets for developing oomycides to control the Phytophthora blight pathogen.

## 4. Materials and Methods

### 4.1. Phytophthora capsici Strains, Bacterial Strains, Culture Conditions, and Test Reagents

*Phytophthora capsici* LT1534, *E. coli* DH5α, and plasmids (pYF515 and pBluescript SK II+) used in the construction of the CRISPR-Cas9 vectors were stored in the laboratory. The mutant ∆*Pctbcc1* was obtained by this assay, which showed that polyethylene glycose (PEG) mediated the protoplast transformation approach [33]. The strain was grown at 25 °C in a V8 medium (10% V8 juice, 0.14% CaCO_3_, 1.5% agar) and used to determine vegetative growth.

### 4.2. Pepper Cultivar

Pepper variety HNUCB0226 was a gift from Prof. Dr. Wang Zhiwei at the College of Horticulture, Hainan University.

### 4.3. Construction of PcTBCC1 Gene Replacement and sgRNA Vectors

To generate *PcTBCC1* gene replacement constructs, genomic DNA extracted from the wild-type (LT1534) strain was used as a template. The upstream (A-fragment) and downstream (B-fragment) flanking sequences of the *PcTBCC1* gene were amplified using *PcTBCC1*-AF/AR and *PcTBCC1*-BF/BR primer pairs listed in Appendix A. The full-length coding region of the hygromycin B phosphotransferase (HPH) *gene* fragment was amplified with the HPH-F/HPH-R primer pair using plasmid pCX62 as a template. A-, HPH-, and B-fragments were cloned into the pBS plasmid. The constructed pBS plasmids were subsequently transformed into competent *E. coli* cells, incubated in resistance-free liquid Luria–Bertani (LB) medium for 60 min, and plated on Luria–Bertani (LB) agar medium containing ampicillin (100 µg·mL^−1^). Positive transformants were screened by colony PCR and confirmed by sequencing.

To construct CRISPR-Cas9 sgRNA cassettes for gene editing, *PcTBCC1*-specific sgRNAs were designed using the web-based Eukaryotic Pathogen CRISPR guide RNA/DNA Design Tool (http://portals.broadinstitute.org/gpp/public/analysis-tools/sgrna-design, accessed on 09 February 2022). sgRNAs targeting positions 101-N-terminal, 590-mid-frame, and 1247-C-terminal were amplified and cloned independently into the pYF515 plasmid, following the procedures described by [33]. Tbcc-sg1-F/Tbcc-sg1-R, Tbcc-sg2-F/Tbcc-sg2-R, and Tbcc-sg3-F/Tbcc-sg3-R primers (Appendix A) were used to synthesize double-stranded sgRNAs targeting positions 29, 450, and 800 of the *PcTBCC1* gene, respectively.

### 4.4. Generation and Identification of PcTBCC1 Gene Deletion Strains

Homogenous mixtures of individual sgRNA constructs of the targeted *PcTBCC1* gene-replacement pBS construct harboring the ligated A-HPH-B frame were transformed into protoplasts prepared from the LT1534 strain using a polyethylene glycol (PEG)-mediated protoplast transformation approach (Appendix A). The transformed protoplasts were cultured in liquid regeneration medium (PM) and incubated under dark conditions in a stationary incubator at 25 °C for 18 h. The resulting regenerated transformants were selected on regeneration agar medium (PAM) co-supplemented with geneticin (G418) 50 µg/mL, hygromycin 100 µg/mL, or both as [34] and incubated at 25 °C to pre-select for G418 resistance. Following 48 h of incubation, V8 juice agar medium supplemented with hygromycin-B was used as the second selection medium and incubated at 25 °C for 3 days. Potentially positive *PcTBCC-1* gene-defective candidates were screened using PCR, RT-qPCR, targeted restricted sequencing, and second-generation whole-genome sequencing (Appendix A).

### 4.5. Total DNA Extraction and Variant Whole Genome Region Analysis

The wild-type LT1534, Δ*Pctbcc1,* and EV (empty vector) strains were cultured in V8 liquid medium at 25 °C and shaken at 120 rpm for 3 days. The hyphae were filtered and rinsed with sterilized ddH_2_O to remove the residual medium, and excess water was drained from the strains using sterilized Whatman filter paper. The processed hyphae were ground in liquid nitrogen to obtain homogeneously powdered hyphae. Equal weights of the samples were weighed into 1.5 mL Eppendorf tubes. The samples were suspended in 900 µL of 2% ^w^/_v_ CTAB buffer (2% ^w^/_v_ CTAB, 8.18 g NaCl, 0.74 g EDTA, 1.21 g Tris-Cl, and 100 mL ddH_2_O), incubated in a 65 °C water bath for 1 h, and centrifuged at 12,000 rpm for 10 min. The 400 µL supernatants were pipetted into sterilized 1.5 mL Eppendorf tubes containing 600 µL chloroform, mixed, and centrifuged at 12,000 rpm for 15 min; this step was repeated twice. The resulting supernatants were pipetted into 2 mL Eppendorf tubes containing 1 mL of absolute ethanol prechilled at −20 °C. Two hundred µL of 3 mol/L NaAc was added, and the contents were transferred into a −80 °C refrigerator for 30 min and then centrifuged for 10 min at 12,000 rpm. The resulting supernatants were discarded, and the precipitates were washed twice with 200 µL of 75% ethanol before centrifuging at 12,000 rpm for 3 min. The alcohol was evaporated through air drying and the fully dried samples were eluted with 50 µL DNase-free water. The extracted DNAs were stored under −20 °C till use. For confirmation of deletion of the full-length *PcTBCC1* gene coding frame in the Δ*Pctbcc1* strains and the TBCC domain deletion strains, the clean reads obtained from the Illumina-based second-generation whole-genome sequencing of the individual strains were mapped to the *P. capsici*-reference genome using Bowtie2 [35] under default parameters. After sorting and indexing using SAMtools 1.17 [36], the aligned results were visualized using IGV [37].

### 4.6. Total RNA Extraction

The wild-type LT1534, Δ*Pctbcc1,* and EV strains were cultured in V8 liquid medium at 25 °C and shaken at 120 rpm for 3 days. The hyphae were filtered and rinsed with sterilized double-distilled water to remove residual medium. After drying the hyphae with sterilized filter paper, they were ground to a powder in liquid nitrogen. Equal weights of each sample were placed in a 1.5 mL sterilized Eppendorf tube, and the mixture was shaken by hand for 30 s to remove proteins. The mixture was left to rest at 25 °C for 3 min, followed by centrifugation at 12,000 rpm for 10 min at 4 °C. In total, 400 µL of the supernatant was transferred into a new 1.5 mL sterilized Eppendorf tube and then 200 µL of absolute ethanol was added, mixed, and added to the adsorption column. After centrifugation at 12,000 rpm for 60 s, the solution was discarded, and 300 L of buffer RW1 was added at 12,000 rpm for 60 s. For the DNase I reaction solution, 75 µL DNase I reaction solution was added to the adsorption column and left to rest for 15 min. Afterward, 500 µL of buffer RW1 was added and the sample was centrifuged at 12,000 rpm for 60 s. The supernatant was discarded, and 500 µL of buffer RW2 was added. Subsequently, the sample was centrifuged at 12,000 rpm for 60 s and the supernatant was discarded. The adsorbent column was removed and placed into a 1.5 mL sterilized Eppendorf tube, 50 µL RNase-free water was added, and the sample was centrifuged at 12,000 rpm for 1 min and then stored at −80 °C until use.

### 4.7. RT-qPCR Verification Assays

To clarify the transcription level of the *PcTBCC1* gene in each deletion mutant, the mutant RNA was reverse-transcribed to obtain cDNA, which was then detected by quantitative real-time PCR (qRT-PCR). The 10 µL reaction mixture was composed of the following: 5 µL premixed SybrGreen, 0.5 µL each of 10-µM forward and reverse primers, 1 µL cDNA template, 2.8 µL RNase-free water, and 0.2 µL ROX II correction solution. qRT-PCR data were generated using QuantStudio Design & Analysis Software (version 1.3.1). Data analysis was performed using the delta Delta-CT ^(2−ΔΔCT)^ method [38]. Tubulin from *Phytophthora capsici* was used as an internal control.

### 4.8. Characterization of Vegetative Growth

Uniform diameters of 6 mm of fresh growing hyphae of the wild type LT1534, ∆*Pctbcc1,* and EV strains were excised from the growing edges of culture plates using 200 µL pipette tips. The excised hyphae plug from the individual strains were inoculated at the center of a 90 mm V8 juice agar culture plate and incubated at 25 °C. The vegetative growth of the individual strains was assessed by measuring the colony diameters at 4 dpi. Five biological growth experiments were performed, each consisting of five technical replicates.

### 4.9. Assessment of Sporangium Production

Uniform hyphae were cut from the edge of the wild type, ∆*Pctbcc1*, and EV strains and inoculated on V8 juice agar medium. The inoculated plates were incubated at 25 °C for 5 d under dark conditions. The Parafilm seals and plates were exposed to 12 h of white light for 3 d to sporangia. The sporangia were washed from individual culture plates and filtered through a Mira-cloth to obtain a 2 mL sporangium suspension. Sporangium quantification was performed by placing 10 µL of the sporangium suspension on a hemocytometer and counting under a microscope. Consistent results were obtained from five biological experiments, each with three technical replicates.

### 4.10. Induction and Assessment of Encystment and the Release of Zoospores

Sporangium suspensions were collected as described above, and the sporangium concentration was adjusted to 2 × 10^4^–4 × 10^4^ mL^−1^. After thorough mixing, 40 µL was placed on hydrophobic microslides. The induction and progression of the pathogenic differentiation of sporangia obtained from individual strains were monitored and counted under a microscope. The observed pathogenic differentiation phenotypes were categorized as non-encyst, zoospore release, pre-encystment abortion, or zoospore burst clots. The encystment and release of cysts/zoospores were induced at 4 °C and monitored at intervals of 0, 30 and 60 min. Consistent results were obtained from five biological experiments, each with three technical replicates.

### 4.11. Pathogenicity Assessment Assay

The individual strains pre-cultured in V8 juice medium for 3 days in a shaking incubator (at a speed of 120 rpm at 25 °C) were filtered and rinsed with sterilized double-deionized water (ddH_2_O). Medium-free mycelia from individual strains were used to independently inoculate leaf tissues of 3–4-week-old seedlings of *P. capsici*-susceptible (HNUCB0226) bell pepper cultivars. The inoculated HNUCB0226 seedlings in the mock group were transferred into a growth chamber, incubated under dark conditions for 24 h, and then exposed to alternating 12 h light/dark conditions. The growth chamber was maintained at 25 °C with a humidity of 75–90%. Disease severity was assessed 72 d post-inoculation. Five biological pathogenicity/virulence experiments (each with five technical replicates) were performed, and 100 seedlings were inoculated per strain.

## Figures and Tables

**Figure 1 ijms-25-12301-f001:**
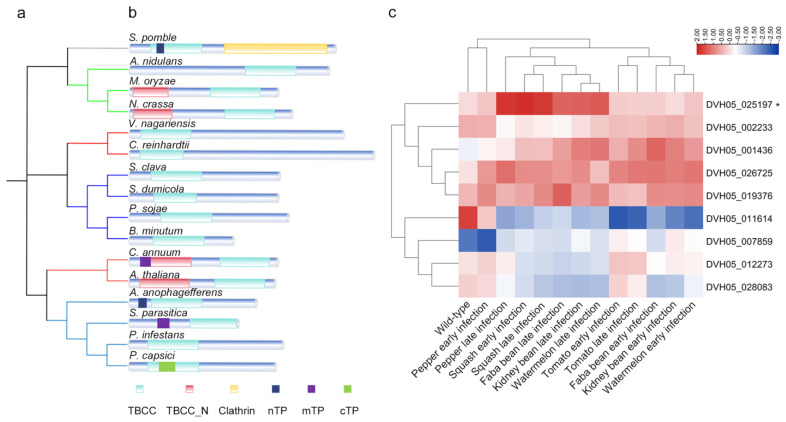
Phylogenetic relationship, domain architecture, and expression pattern of *PcTBCC1* during *P. capsici*–pepper interaction. (**a**) The cladogram shows the prevailing phylogenetic relationship between TBCC domain-containing proteins from selected organisms. MEGA X was used to construct the neighbor-joining trees. (**b**) The trees represent the comparative structure and distribution of the TBCC domain in TBCC domain-containing proteins identified in the selected organisms. (**c**) The heatmap represents the expression of the individual genes coding for TBCC domain-containing proteins identified in *P. capsici* at early (8–24 hpi) and late (36–72 hpi) stages of pathogen–host interaction. The RNA sequencing data were obtained using a deep-sequencing approach, and the individual expression patterns were consistent in all seven replicates. The gene expression level was computed using the RFPKM method. * highlight the position of PcTbcc in the phylogenetic tree and on the Heatmap.

**Figure 2 ijms-25-12301-f002:**
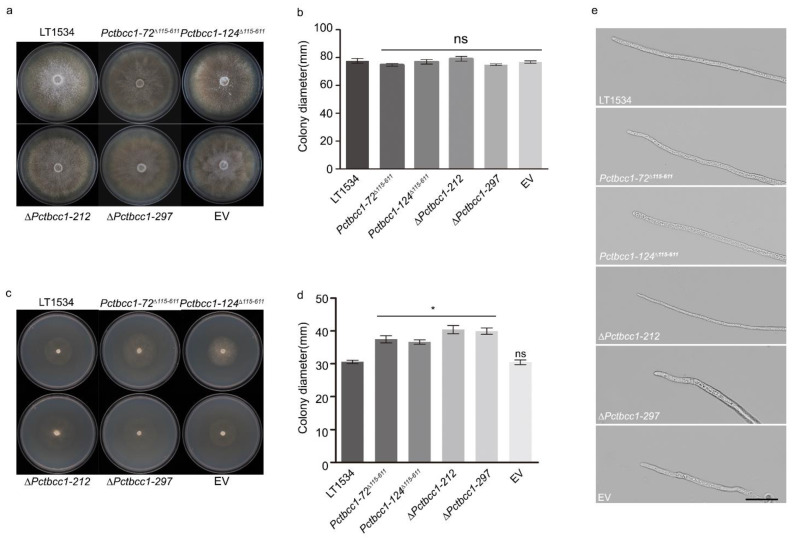
Contributions of PcTbcc1 to vegetative development and hypha morphogenesis under sufficient and deficient nutrient conditions. (**a**) Comparative vegetative growth performances of the wild-type, *∆Pctbcc1-212, ∆Pctbcc1-297*, *Pctbcc1-72^∆115-611^*, *Pctbcc1-124^∆115-611^^1^*_,_ and EV strains on nutrient-sufficient V8 juice agar medium. (**b**) Statistical computation of average colony diameters for the individual strains grown on nutrient-sufficient V8 juice agar medium for 4 days. (**c**) Comparative vegetative growth performance of the individual strains on a nutrient-deficient medium. (**d**) Average colony diameters for the individual strains. Statistical analyses were performed with consistent results obtained from five biological replicates, each with five technical replicates. Error bars represent the standard deviation between replicates, while single (“*”) asterisks represent a statistically significant difference (*p* ≤ 0.05), and ns: no significant difference. Scale bar = 10 μm. (**e**) Micrographs showing the morphology of vegetative hyphae produced by the individual strains. Statistical significance was analyzed using a one-way ANOVA (non-parametric) approach.

**Figure 3 ijms-25-12301-f003:**
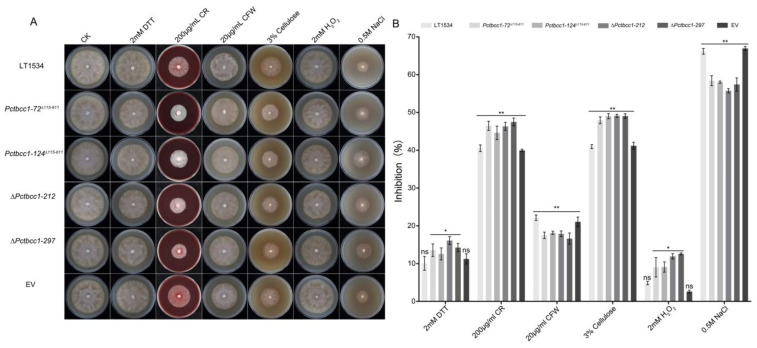
The impacts of targeted replacement of the PcTBCC1 gene on the tolerance of *P. capsici* to different stress-inducing osmolytes during vegetative development. (**A**) Vegetative growth characteristics of the wild-type, ∆*Pctbcc1-212*, ∆*Pctbcc1-297*, *Pctbcc1-72*^∆*115-611*^, and *Pctbcc1-124*^∆*115-61*^*^1^*, and EV strains on V8 juice agar medium supplemented independently with 2 mM DTT, 200 μg/mL CR, 20 μg/mL CFW, 3% ^w^_/v_ cellulose, 2 mM H_2_O_2_, and 0.5 M NaCl. (**B**) Statistical presentation of the effects of individual stress-inducing osmolytes on the vegetative development of the individual strains. Statistical analyses were performed with consistent results obtained from five biological replicates, each with five technical replicates. Inhibition rate = (the diameter of untreated strain—the diameter of treated strain)/(the diameter of untreated strain) × 100%. Error bars represent the standard deviation between replicates, while single (“*”) and double (“**”) asterisks represent a statistically significant difference of *p* ≤ 0.05 and *p* ≤ 0.01, respectively. Statistical significance was analyzed using a one-way ANOVA (non-parametric) approach. ns: no significant difference.

**Figure 4 ijms-25-12301-f004:**
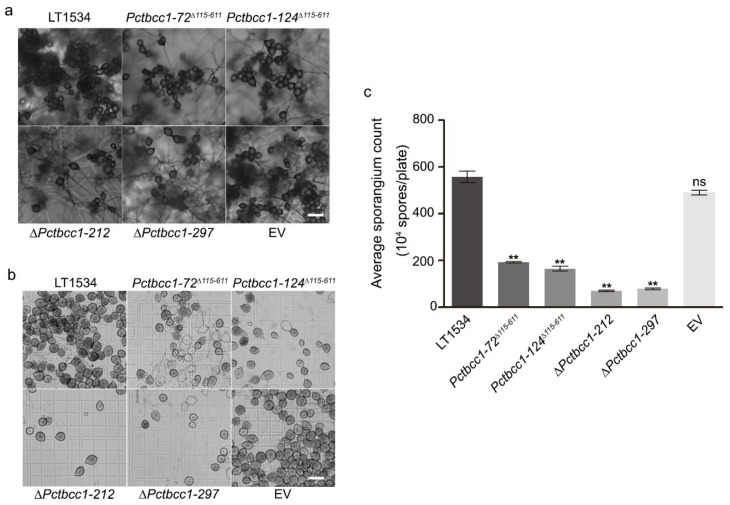
Targeted replacement of the *PcTBCC1* gene significantly attenuated asexual sporulation in *P. capsici*. (**a**,**b**) Micrograph showing the comparative asexual sporulation characteristics of the individual strains incubated under optimum asexual sporulation conditions for 3 days. Scale bar = 20 μm. (**c**) Comparative statistical quantification of sporangium production in the wild-type, ∆*Pctbcc1-212*, ∆*Pctbcc1-297*, *Pctbcc1-72*^∆*115-611*^*, Pctbcc1-124*^∆*115-61*^, and EV strains. The results were statistically analyzed using data from five independent biological experiments, with five technical replicates. The error bars represent standard deviation, while double (“**”) asterisks represent a significant difference of *p* ≤ 0.01. Statistical significance was analyzed using a one-way ANOVA (non-parametric) approach. ns: no significant difference.

**Figure 5 ijms-25-12301-f005:**
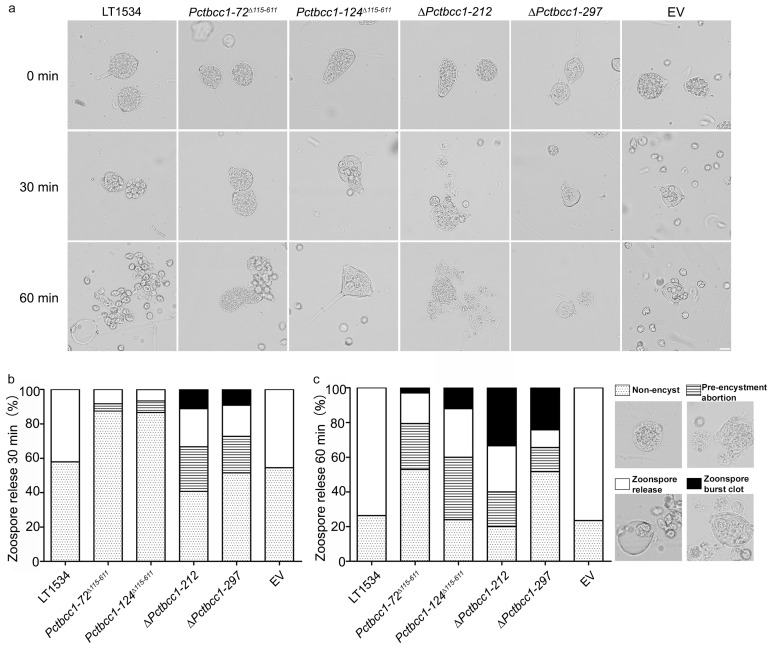
Influence of PcTbcc1 dysfunctions on encystment, zoospore release, and survival in *P. capsici*. (**a**) Micrograph showing the sporangium morphology, encystment, release, and survival of wild-type, *∆Pctbcc1-212*, *∆Pctbcc1-297*, *Pctbcc1-72^∆115-611^*, *Pctbcc1-124^∆115-611^*, and EV strain zoospores at different stages of pathogenic differentiation in *P. capsici*. Scale bar = 20 μm. (**b**,**c**) Comparative statistical quantification of the encystment, release, and survival of zoospores produced by the individual strains. One hundred sporangia per strain were monitored in each technical replicate. Consistent results from five independent biological experiments (each consisting of three technical replicates; the total number of sporangia (n) counted is given as (n = 100 × 5).

**Figure 6 ijms-25-12301-f006:**
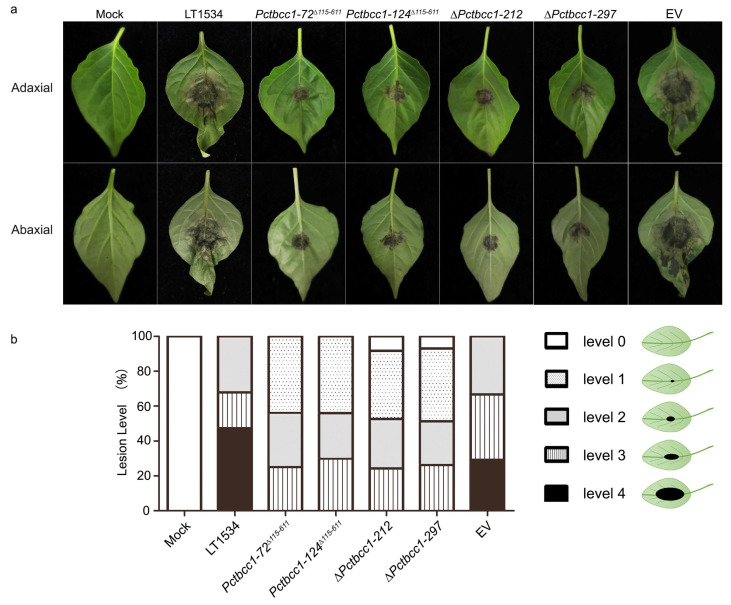
PcTbcc1 is essential for the full virulence of *P. capsici.* (**a**) Virulence characteristics of the wild-type, *∆Pctbcc1-212*, *∆Pctbcc1-297*, *Pctbcc1-72^∆115-611^*, *Pctbcc1-124^∆115-61^^1^*, and EV strains against 4-week-old *P. capsici*-susceptible HNUCB0226 bell pepper seedlings at 72 hpi. (**b**) Statistical computation of disease severity index through the quantification and scoring of the different types of blight lesions on leaf tissues of pepper seedlings independently inoculated with individual strains after 72 hpi. Consistent results from five independent biological experiments, each with three technical replicates, were used for statistical analyses. Statistical significance was analyzed using a one-way ANOVA (non-parametric) approach. In each biological experiment, we inoculated 20 seedlings per strain. Therefore, the total number of seedlings (n) per strain is n = 20 × 5.

**Figure 7 ijms-25-12301-f007:**
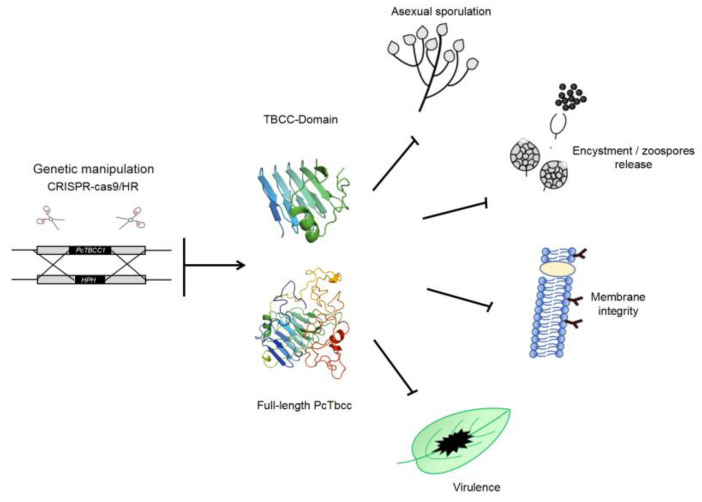
Model of tubulin-binding cofactor C-mediated regulation of nutrient-dependent signaling developmental events in *P. capsici*. In response to nutrient limitation, PcTbcc1 functions as a genetic switch that coordinates the microtubule- and membrane-dependent transduction of signals to facilitate the transition of *P. capsici* from the vegetative growth phase to the asexual reproduction phase.

## Data Availability

The datasets used and/or analyzed during the current study are available from the corresponding author on reasonable request. The whole genome sequencing of wild-type and gene-deficient strains of *Phytophthora capsici* has been deposited in the Sequence Read Archive. BioProject ID: PRJNA1070426 with Bio-Sample numbers SRX23576275, SAMN39654075, SAMN39654074, and SAMN39652131.

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
