# Peer review of "TBCC Domain-Containing Protein Regulates Sporulation and Virulence of Phytophthora capsici via Nutrient-Responsive Signaling"

_ijms, 2024, doi:10.3390/ijms252212301_

Round 1

Reviewer 1 Report

Comments and Suggestions for Authors

I was invited to review the article "TBCC domain-containing protein regulates sporulation and virulence of Phytophthora capsici via nutrient-responsive signaling". 

The article describes a funcional analysis of TBCC proteins in pathogenesis of an important phytopathogen as P. capsici. The data seem sound and original. However, I have some questions and critics about this manuscript:

Major concerns:

- The figure legends must be modified. They are not descriptive and have a series of problems in grammar and written style.

- Data of chemical stress are not clear for me. Changes in phenotype and mycelial growth are not so evident. Please, re-reavaluate this data or provide better images and more convincing statistical data. 

Minor concerns:

- Line 106: The description of reagents seems unnecessary

- Please, review references citations in the middle of the text and in reference list. 

- Figure 7 may be improved since it not clearly connects protein function with mechanisms interference after mutant production. 

Comments on the Quality of English Language

English must be reviewed. In my opinion, the written style is prolixous and must be evaluated by a reviewer. 

Author Response

Reviewer summary comment: The article describes a funcional analysis of TBCC proteins in pathogenesis of an important phytopathogen as P. capsici. The data seem sound and original. However, I have some questions and critics about this manuscript:

Authors Summary response: The reviewers summary accurately reflects the content of our manuscript.

 Reviewer comment-1: The figure legends must be modified. They are not descriptive and have a series of problems in grammar and written style.

Authors response-1: We revised and improve the figure legend in our revised manuscript

Reviewer comment-2: Data of chemical stress are not clear for me. Changes in phenotype and mycelial growth are not so evident. Please, re-evaluate this data or provide better images and more convincing statistical data. 

Authors response-2: We apologize for not providing the formulae for computing the effects of different chemicals on the individual strains. The inhibition effects of different chemicals was rigorously computed using consistent results from five biological replicates, each with five technical replicates using the formula’; Inhibition rate = (the diameter of untreated strain – the diameter of treated strain)/(the diameter of untreated strain) × 100%. We have revised and updated the figure in the revised manuscript.

Reviewer comment-3: Line 106: The description of reagents seems unnecessary

Authors response-3: Correction effected according to recommendation

Reviewer comment-4: Please, review references citations in the middle of the text and in reference list. 

Authors response-4: Correction effected according to recommendation

Reviewer comment-5: Figure 7 may be improved since it not clearly connects protein function with mechanisms interference after mutant production.

Authors response-5: As interpreted figure legend of Figure 7, we believe that the current model showed the deletion of PcTBCC gene which directly relates to loss of protein function compromised morphogenesis, reproduction, stress tolerance and virulence in P. capsici. We believe there is ample connection between protein and and the phenotypic defects observed displayed by the PcTBCC gene deletion strains.

 Reviewer comment-6: English must be reviewed. In my opinion, the written style is prolixous and must be evaluated by a reviewer. 

Authors response-6: Thanks for the suggestion, was revised my native English speakers and reviewed again by additional English professors to eliminate or minimize the prolixity.

Reviewer 2 Report

Comments and Suggestions for Authors

Journal: Int. J. Mol. Sci.-3258776

Ms. Title: TBCC domain-containing protein regulates sporulation and virulence of Phytophtora capsica via nutrient-responsive signaling.

The manuscript authorized by Guo et al. deals with an important matter concerning phytopathogenic research and further application to increase the production of crops with essential nutritional roles, namely the use of advanced technology (CRISPR-Cas9) to unravel new molecular mechanisms involved in the TBCC-domains on the blight disease caused Phytophtora capsici. In general terms, the experimental design and goals are correct and the results consistent with the proposed hypothesis. The text is also well-written. However, the manuscript should not be accepted until some important questions are properly addressed. Please, pay attention to the following items:

-The description in Methods of the molecular biology procedures is detailed and exhaustive, It should be condensed. In contrast, some relevant information concerning other methods is scarce. Thus, some culture media are customary in lab. (i.e. LB medium), but the reports on PEG-mediated protoplasts or the V8 media are poor. Furthermore, the distinction between wild-type and pathogenic phenotypes or the assays for pathogenicity are insufficiently explained.

-In the same way, the proposed differential roles of the nine TBC domains in host-pathogen interactions require any conceptual support.

-My main query comes from the fact that some results are reported in a doubtful manner (…we infer, we speculate…). Two points should be addressed here: (i) I am not sure the mutants obtained after genetic constructions are adequate for the goals proposed; (ii) The data obtained from physiological analysis (vegetative, sporulation, zoospores/cyst formation) show fluctuations between mutant and wild-type strains. I wonder about the effects of a null mutant as well as a recommendation of precise control assays.

Othe questions

-Italics phrases are introduced inadequately in some parts of the text (see the Title).

-The distinction between pathogenicity and virulence (l. 264) needs to be simplified.

-The terminology of mutants is very complex.

-Although the text is well-written, a general survey will improve the final presentation.

Author Response

Reviewer 2

Reviewer summary comment: The manuscript authorized by Guo et al. deals with an important matter concerning phytopathogenic research and further application to increase the production of crops with essential nutritional roles, namely the use of advanced technology (CRISPR-Cas9) to unravel new molecular mechanisms involved in the TBCC-domains on the blight disease caused Phytophtora capsici. In general terms, the experimental design and goals are correct and the results consistent with the proposed hypothesis. The text is also well-written. However, the manuscript should not be accepted until some important questions are properly addressed. Please, pay attention to the following items:

Authors Summary response: The reviewers summary accurately reflects the content of our manuscript.

Reviewer comment-1: The description in Methods of the molecular biology procedures is detailed and exhaustive, It should be condensed. In contrast, some relevant information concerning other methods is scarce. Thus, some culture media are customary in lab. (i.e. LB medium), but the reports on PEG-mediated protoplasts or the V8 media are poor. Furthermore, the distinction between wild-type and pathogenic phenotypes or the assays for pathogenicity are insufficiently explained.

Authors response-1: We are grateful for the valuable comments and suggestions. Some of the molecular biology procedures used our study comes with some slight modifications therefore, we thought it was appropriate to provide detail descriptions to serve as a blue-print for other researchers who might find it interesting to adopt or modify further. Since, PEG-mediated protoplast transformation in oomycete and fungi, we humbly wish to repeat the details as it have been extensively  covered in literature and we have cited some of the literature in support. We have revised the manuscript to include details with regards to the formulation of V8 and LB media.

Reviewer comment-2: In the same way, the proposed differential roles of the nine TBC domains in host-pathogen interactions require any conceptual support.

Authors response-2: We are equally interested in unique roles of the individual TBC protein-coding genes in P. capsici but conceptualization of nine proteins that are deemed significantly important in the progression of vital cellular processes could academically be misleading, especially since we have initiated processes to functionally characterized the remaining TBC protein-coding genes in P. capsici, We should be in position to provide a comprehensive conceptual framework direct/indirect roles of the TBC domain-containing proteins in pathophysiological development of P. capsici.

Reviewer comment-3: My main query comes from the fact that some results are reported in a doubtful manner (…we infer, we speculate…). Two points should be addressed here: (i) I am not sure the mutants obtained after genetic constructions are adequate for the goals proposed; (ii) The data obtained from physiological analysis (vegetative, sporulation, zoospores/cyst formation) show fluctuations between mutant and wild-type strains. I wonder about the effects of a null mutant as well as a recommendation of precise control assays.

Authors response-3: We appreciate the insightful query, Our initial positions were based on the fact that tubulin-associated processes regulates the progression of multiple vital cellular and physiological events. We reason that genetic manipulations and fundamental phenotypic characterization only serve to provide a solid foundation further extensive proteomic, metabolomics and interactome evaluation and thought it was appropriate to hedge our conclusion with the use of non-categorical diction like “infer, or speculate” to emphasize that the reported phenotype could be either a direct or indirect manifestations of the dysfunction of TBCC in P. capsici but since this seems to have caste some forms doubt over the stability of the targeted PcTBCC gene replacement or edited mutants strains, we have accordingly revised the choice of words in our revised munuscript. Results from PCR, qPCR, and genome sequencing results, and the computed degree of error recorded between technical and biological replicates is an ample confirmation of the consistency in the phenotypic defects associated with mutant strains and consistency in differences between mutant strains and the wild-type.  

Reviewer comment-4: Italics phrases are introduced inadequately in some parts of the text (see the Title).

Authors response-4: We have taken note of this crucial observations and have corrected them accordingly

Reviewer comment-5: The distinction between pathogenicity and virulence (l. 264) needs to be simplified.

Authors comment-5: Pathogenicity as applied in plant pathology refers to ability of a potential microbial pathogen to penetrate host cells or tissues while virulence ( previously termed as aggressiveness) on the other describes the ability or the rate at which pathogens to proliferate host tissues.

Reviewer comment-6: The terminology of mutants is very complex.

Authors response-6: That is correct but the use of mutants within context of this manuscript only refers targeted PcTBCC gene replacement or edited strains without ambiguity.

Reviewer comment-7:-Although the text is well-written, a general survey will improve the final presentation.

Authors response-6: Thanks for the encouragement, we have revised some diction and phrase in the revised manuscript.

Round 2

Reviewer 2 Report

Comments and Suggestions for Authors

The revised version of this manuscript has now been evaluated.

In general terms, the authors are receptive to my queries and have addressed the majority in a correct way, although I would still have some punctual minor discrepancies. However, I am certainly reluctant to the extensive and too dense description of the methodology employed here.

I am not a deep expert in this specific topic, but I have some notions, and I am truly convinced that a clearer report of this section would help to a better understanding of the whole picture. It does not seem a difficult task.